

# Nest-site selection and breeding success of passerines in the world's southernmost forests

Rocío Fernanda Jara[1,2,3], Ramiro Daniel Crego[2,3,4], Michael David Samuel[5], Ricardo Rozzi[2,3,6,7,8] and Jaime Enrique Jiménez[1,8]

[1] Department of Biological Sciences, University of North Texas, Denton, TX, United States of America
[2] Sub-Antarctic Biocultural Conservation Program, University of North Texas, Denton, TX, United States of America
[3] Institute of Ecology and Biodiversity, Omora Park Field Station, Puerto Williams, Chile
[4] Conservation Ecology Center, National Zoological Park, Smithsonian Conservation Biology Institute, Front Royal, United States of America
[5] Department of Forest and Wildlife Ecology, University of Wisconsin-Madison, Madison, WI, United States of America
[6] Universidad de Magallanes, Punta Arenas, Chile
[7] Department of Philosophy and Religion, University of North Texas, Denton, TX, United States of America
[8] Advanced Environmental Research Institute, Department of Biological Sciences, University of North Texas, Denton, TX, United States of America

Corresponding author
Rocío Fernanda Jara,
RocioJara@my.unt.edu

## ABSTRACT

**Background**. Birds can maximize their reproductive success through careful selection of nest-sites. The 'total-foliage' hypothesis predicts that nests concealed in vegetation should have higher survival. We propose an additional hypothesis, the 'predator proximity' hypothesis, which states that nests placed farther from predators would have higher survival. We examined these hypotheses in the world's southernmost forests of Navarino Island, in the Cape Horn Biosphere reserve, Chile (55°S). This island has been free of mammalian ground predators until recently, and forest passerines have been subject to depredation only by diurnal and nocturnal raptors.

**Methods**. During three breeding seasons (2014–2017), we monitored 104 nests for the five most abundant open-cup forest-dwelling passerines (*Elaenia albiceps*, *Zonotrichia capensis*, *Phrygilus patagonicus*, *Turdus falcklandii*, and *Anairetes parulus*). We identified nest predators using camera traps and assessed whether habitat characteristics affected nest-site selection and survival.

**Results**. Nest predation was the main cause of nest failure (71% of failed nests). *Milvago chimango* was the most common predator, depredating 13 (87%) of the 15 nests where we could identify a predator. By contrast, the recently introduced mammal *Neovison vison*, the only ground predator, depredated one nest (7%). Species selected nest-sites with more understory cover and taller understory, which according to the total-foliage hypothesis would provide more concealment against both avian and mammal predators. However, these variables negatively influenced nest survival. The apparent disconnect between selecting nest-sites to avoid predation and the actual risk of predation could be due to recent changes in the predator assemblage driven by an increased abundance of native *M. chimango* associated with urban development, and/or the introduction of exotic mammalian ground predators to this island. These predator assemblage changes could have resulted in an ecological trap. Further research will be

needed to assess hypotheses that could explain this mismatch between nest-site selection and nest survival.

## INTRODUCTION

Where do birds place their nests? This question has intrigued ornithologists since the early days of the discipline (*Birkhead, Wimpenny & Montgomerie, 2014*; *Lovette & Fitzpatrick, 2016*). For open-cup nesters, early studies pointed to food availability as the most important factor for nest-site selection, but predation has been increasingly considered as another major factor (*Martin, 1987*; *Martin, 1993*; *Reidy & Thompson, 2018*). Predation can directly affect survival of eggs, juveniles, and adults, and has been identified as the main cause of nest failure in passerines (*Nice, 1957*; *Ricklefs, 1969*; *Liebezeit & George, 2002*; *Bellamy et al., 2018*; *Reidy & Thompson, 2018*). According to these studies, we predict that birds will select those habitat characteristics that reduce predation risk and thus increase the probabilities of nest survival (*Jaenike & Holt, 1991*; *Fontaine & Martin, 2006*).

Several hypotheses have been proposed to explain the mechanisms by which nest placement reduces predation. One of these, the 'total-foliage' hypothesis, predicts that nests located in sites with more surrounding foliage would have higher concealment, as well as more interference with the transmission of odors and sounds that could be detected by a predator. Thus, a larger amount of foliage reduces predation risk (*Martin & Roper, 1988*; *Martin, 1993*). In the present study we introduce another, but not mutually exclusive hypothesis, which we call the 'predator proximity' hypothesis. This hypothesis assesses types of predators according to their mode of attack, particularly aerial versus terrestrial. This hypothesis assumes that passerine birds select nest sites that avoid discovery and attack by the major type of predators in their ecosystem, and it predicts that: (i) when predation is dominated by aerial predators, birds will place nests near the ground and (ii), in contrast, when predation is dominated by ground predators, birds will place nests at greater height from the ground (*Jara et al., 2019*). Another factor that we consider in this hypothesis is canopy cover. Some aerial predators search for prey while perched in the canopy. Hence in habitats dominated by aerial predators that exhibit sit and wait behavior, we predict that passerine birds will place nests in sites where there is less canopy cover and/or where the canopy is taller (both factors, will effectively put raptors farther away from nests placed in the understory).

High-latitude forests offer ideal natural laboratories because they have a simpler structure compared to tropical forests (i.e., the canopy is dominated by a few species belonging to only one genus, and the understory has low abundance and richness of shrub species; *Rozzi et al., 2008*). Consequently, sub-Antarctic forests of South America provide unique opportunities to test the total-foliage and predator proximity hypotheses and collect evidence to understand the mechanisms that explain nest-site selection and nest survival.

Navarino Island (55°S), located in the Cape Horn Biosphere Reserve, hosts the world's southernmost forests (*Rozzi et al., 2012*) and serves as the breeding ground to 28 bird species (*Ippi et al., 2009*; *Rozzi, 2010*). Here, passerines are the most diverse and abundant group of terrestrial vertebrates, due to the absence of herpetofauna and the limited number of native terrestrial mammals (*Dardanelli et al., 2014*). Hence, nest-site selection takes place in the context of a simple assemblage of vertebrate predators, which until the end of the twentieth century included only diurnal and nocturnal raptors (e.g., *Accipiter chilensis*, *Caracara plancus*, *Glacidium nana*, *Falco sparverius*, *Milvago chimango*, and *Strix rufipes*; *Ippi et al., 2009*; *Schüttler et al., 2009*). Among the most common open-cup passerines breeding in these forests are the White-crested Elaenia (*Elaenia albiceps*), Rufous-collared Sparrow (*Zonotrichia capensis*), Patagonian Sierra-Finch (*Phrygilus patagonicus*), Austral Thrush (*Turdus falcklandii*), and Tufted Tit-Tyrant (*Anairetes parulus*) (*Rozzi & Jiménez, 2014*). Although abundant across their range (*Medrano et al., 2018*), little is known about these species regarding their nesting habits and nest survival.

In other systems birds prefer to nest in sites with lower risk of depredation by avian predators (*Sergio, Marchesi & Pedrini, 2003*; *Roos & Pärt, 2004*; *Latif, Heath & Rotenberry, 2012*). On Navarino Island, bird nesting strategies also may be aimed at reducing the risk of depredation by raptors, the top native predators in this ecosystem. Preliminary evidence suggests that, for example, *T. falcklandii* on Navarino Island breeds closer to the ground than mainland populations (*Jara et al., 2019*) where the predator assemblage includes several terrestrial species such as wild cats and foxes (*Zúñiga, Muñoz Pedreros & Fierro, 2008*; *Altamirano et al., 2013*). However, the simple predator–prey system on Navarino Island, dominated almost exclusively by raptors, was disrupted two decades ago with the introduction of the American mink (*Neovison vison*) (*Rozzi & Sherriffs, 2003*), and the rapid increase of feral domestic cats (*Felis catus*) and dogs (*Canis lupus familiaris*) (*Rozzi et al., 2006b*). These three exotic predators actively prey on passerine birds on Navarino Island (*Schüttler, Cárcamo & Rozzi, 2008*; *Schüttler, Saavedra-Aracena & Jiménez, 2018*), and worldwide (*Ferreras & Macdonald, 1999*; *Bartoszewicz & Zalewski, 2003*; *Doherty et al., 2016*). Hence, the arrival of these mammals presented a new predation pressure for birds nesting on Navarino Island and may represent an ecological trap for birds that evolved in the absence of terrestrial predators. The increasing abundance of these novel predators during the first two decades of the 21st century coincides with the rapid disappearance from the island of the Magellanic Tapaculo (*Scytalopus magellanicus*), a small passerine with poor flying capacity that inhabits the understory of South American temperate forests (*Rozzi et al., 1996*). This bird was detected in the Omora Ethnobotanical Park until 2003 (*Ippi et al., 2009*), but not in recent surveys of the area (*Rozzi & Jiménez, 2014*, RD Crego, pers. comm., 2015).

According to the total-foliage hypothesis, to reduce the risk of predation, passerines should nest in sites that provide more nest concealment (Table 1). According to the predator proximity hypothesis, passerines should select nest-sites that avoid the presence of predators, thus reducing the risk of predation. Based on these hypotheses, we predicted that on Navarino Island birds place nests in sites with denser and taller understory, and would avoid placing nests close to the canopy (exposing them to perched raptors), or too

**Table 1** Variables incorporated into candidate models assessing habitat characteristics influencing nest-site selection by five forest passerines on Navarino Island, Chile.

| Variable | Hypothesis | Predictions | Rationale |
|---|---|---|---|
| Canopy cover | Predator proximity | Negatively associated with nest presence | More canopy cover allows for the presence of aerial predators, imposing a threat to nesting birds and their nests. |
| Canopy height | Predator proximity | Positively associated with nest presence | Higher canopy puts perched raptors farther away from birds nesting in the understory, making it harder to detect bird breeding activity. |
| Understory cover | Total-foliage | Positively associated with nest presence | More understory cover provides more visual nest concealment and interferes with the transmission of odors and sounds coming from the nest that could be detected by a predator. |
| Understory height | Predator proximity/Total-foliage | Positively associated with nest presence | Taller understory provides more nest concealment against predators, and allows for higher nest placement, which reduces accessibility for ground predators. |

close to the ground (exposing them to recently introduced ground predators) (Table 1). We also predicted that survival rates would be lower in nests located at these extremes of the vertical axis of the forest structure. To test these hypotheses, we collected data on forest-dwelling passerines in the world's southernmost forests with two general goals: (i) to test the importance of habitat characteristics on nest-site selection, and (ii) to determine how habitat characteristics and temporal variables influence daily nest survival rate (DSR). We examined habitat variables that are relevant for nest survival according to the total-foliage and predator proximity hypotheses (Table 2).

## MATERIALS & METHODS

### Study site

The study site is located on the northern coast of Navarino Island (54°S), within the Cape Horn Biosphere Reserve, at the southern end of South America. Its forests encompass a mixture of only six tree species, and are dominated by the broadleaf evergreen species *Nothofagus betuloides* (*Rozzi et al., 2008*). The understory has low abundance and diversity of shrub species, but is covered by s diverse and dense carpet of bryophytes (*Rozzi et al., 2008*). The regional climate is oceanic, resulting in a mean rainfall of 467 mm homogeneously distributed throughout the year, and in low annual temperature range, with a mean temperature of 10.8 °C during warmest month in summer of and 1.9 °C in the coldest month in winter (*Rozzi et al., 2014*). We surveyed for nests along 28 km throughout the northern shore forests; however, most of our efforts were concentrated within the more accessible and protected forests in the Omora Ethnobotanical Park (54°56′S, 67°39′W) (*Rozzi et al., 2006a*).

### Nest searching and monitoring

We searched for nests during three breeding seasons: 2014–2015 (November–January), 2015–2016 (October–February) and 2016–2017 (October–January). We located active nests (under construction or containing at least one egg or young) by observing and following adults exhibiting signs of breeding or nesting behavior (carrying nest material,

Table 2 Justification of variables incorporated into candidate models for estimating daily nest survival rate of five forest passerines on Navarino Island, Chile.

| Model | Variable | Predictions | Rationale |
|---|---|---|---|
| Null | Intercept only | Nest survival is random | Assumes daily survival rate (DSR) is constant. |
| Temporal effects | Day of year | Negatively associated with DSR | Late nesters will have lower nest survival because of the overlap with increased depredation pressure in the forest interior (i.e., *N. vison*), due to their breeding dynamics. |
| | Nest age (linear vs quadratic effects) and nest stage | Negatively associated with DSR | Nest age and stage influence adult behavior around the nest (increased nest visitation for food provisioning), and increased noise and odor from nestlings. These cues could be detected by predators. |
| Habitat effects | Concealment | Positively associated with DSR | Under the 'total-foliage' hypothesis, more nest concealment not only protects the nest and its content from predators, but also the adults entering and leaving it. |
| | Nest height off the ground (linear vs quadratic effects) | Positively associated with DSR | Under the 'predator proximity' hypothesis, nests closer to the ground will be more susceptible to ground predators |
| | Ground predator index | Negatively associated with DSR | Under the 'predator proximity' hypothesis, nests with higher index score will be more susceptible to predation. |
| | Canopy cover, canopy height, understory cover and understory height | Variables associated with nest-site selection will have equivalent effect on DSR. | Rationale of these variables' effect on DSR is equivalent to that described in nest-site selection (Table 1). |

defending territory via alarm call, or carrying food or fecal sacs in their bills). In cases where we suspected the nest was in a well-delineated small area, but we were unable to see it, we scanned the vegetation with a thermal imaging camera (FLIR One, 2014© FLIR® Systems, Inc.) to help locate the nest. We monitored active nests until young fledged or the nest failed, using both camera traps (Bushnell Trophy Cam: Bushnell Corp., Overland Park, KS, USA) and nest visitation. We deployed a camera trap between 1–3 m from the nest, depending on the surrounding vegetation. We set cameras to take three consecutive pictures per trigger (to increase chances of detecting the predator) and set a minute delay between triggers. We did not deploy cameras during the laying and early incubation period to prevent nest abandonment (*Pietz & Granfors, 2000*). Approximately 10% of nests did not have cameras deployed at any stage. We typically visited nests every other day, unless we suspected a possible change of nest developmental stage (i.e., laying, incubation, nestling), in which case we visited them every day. During our nest visits, we verified that no predators were in the vicinity to observe our movements and later prey on the nest. Otherwise, we did not approach the nest at that time. We considered a nest successful if: (i) the nest was empty and there were fledglings near it, (ii) the camera detected them fledging in the absence of predators, and/or (iii) the nest was empty and there was fecal matter on the rim of the nest or underneath it. We considered a nest to have failed if: (i) there were dead nestlings on or around it, (ii) it was empty (either intact or destroyed) before the earliest possible date of fledging, or (iii) the eggs never hatched and there was no adult activity (i.e., abandoned during incubation).

## Nest site characteristics

After nesting ended, we characterized the nest site following a modified BBIRD protocol (*Martin et al., 1997*). We measured habitat features that might influence the presence of predators and their ability to find nests, including potential perching substrates for raptors, and features that contribute to nest concealment. Within a 5-m radius plot, centered on the nest, we recorded nest height from ground (cm) (hereafter nest height, measured to the rim of the nest), mean nest coverage (%) (hereafter, concealment, estimated as the mean nest coverage measured from 1 m above the nest and from each cardinal direction), canopy cover (%), canopy height (m), understory cover (%), and understory height (cm). We also visually estimated a ground predator (i.e., American mink, rodents, dogs, and cats) accessibility index for every nest. This index ranged from 0–2 with 0 indicating nests that were difficult for a ground predator to access (i.e., nest placed high in a tree without easily accessible branches from the ground), 1 indicating nests that could be accessible from the ground (i.e., nest above ground level but of easy access for a ground predator through climbable branches), and 2 indicating nests that were placed on the ground and could have been easily accessed by potential ground predators.

We assessed nest site selection by measuring the same habitat characteristics (except those specifically related to the nest) using a paired-random plot for each nest. Each random plot was located at a random direction and random distance between 25–70 m from the nest. We chose this distance to maximize the chances the plot was within the home range of the breeding pair. However, because there is no information of home range sizes for these species, these distances are based on personal observations during the study. Before we measured habitat characteristics at the random-paired plot, we verified that active nests of these species were not present at the plot.

## Statistical analyses
### Nest-site selection

We used logistic regression to investigate whether habitat characteristics influenced nest-site selection. We developed separate candidate models for each species to assess the probability that a plot contained a nest as a function of canopy cover, canopy height, understory cover, and understory height (Table 1). The response variable was either 1 or 0, indicating presence or absence of a nest, respectively. We ran these four univariate models, as well as all possible combinations of variables, excluding interactions, and estimated their Akaike information criterion corrected for small sample size ($AIC_c$) (*Burnham & Anderson, 2002*). We selected the top model as the one having the lowest $AIC_c$, and evaluated parameter importance by determining whether or not their 95% confidence interval (CI) included zero (*Tabachnick & Fidell, 2001*). Before fitting the models, we checked for outliers with Cook's distance (D), and for correlation among covariates ($r > 0.75$). For *T. falcklandii* there was one outlier for understory height (Cook's $D > 1$). Replacing this value with the mean of the variable produced similar results as the original value. Furthermore, this variable did not have a meaningful effect on the response variable (see 'Results'). Therefore, we conducted the analysis with this outlier in the data. We used $\chi^2$ tests to determine goodness of fit of the final models, accepting the model if $p > 0.05$. We calculated the odds ratio to determine

the effect of significant habitat predictor variables on the likelihood of a plot containing a nest.

### Nest survival

We used the logistic exposure method (*Shaffer, 2004*) to investigate temporal and habitat variables that influenced daily nest survival rate (DSR) by species (Table 2). We evaluated alternative models using a two-stage process. First, we evaluated temporal variables: nest age (days since first egg was laid; linear vs quadratic effects), nest stage (egg [laying and incubation] vs nestling), and day of year (linear vs quadratic effects). We used the best model from this first stage (the one with lowest $AIC_c$) as the starting model and evaluated habitat variables in the second stage: concealment, canopy cover, canopy height, understory height, understory density, nest height (linear vs quadratic effects), and ground predator accessibility index. From the second stage, we selected the model with lowest $AIC_c$ as the final model for each species. We evaluated the importance of each parameter in the final model by determining whether their 95% CI included zero (*Tabachnick & Fidell, 2001*). For both stages, we built candidate models using all possible combinations of variables, excluding interactions. Finally, we assessed the goodness of fit of the final models with $\chi^2$ tests, accepting the model if $p > 0.05$. We estimated overall nest survival with the final DSR model for every species, holding continuous variables at their standardized mean value ($\bar{x} = 0$). For models with categorical variables, we estimated a separate DSR for each level of the variable(s). To estimate total survival, we raised DSR to an exponent equal to the average number of risk days (i.e., either per nesting stage or whole nesting cycle) per species. We used duration of incubation and nestling periods determined for these species in the same study area (*Jara et al., 2019*). Because the duration of incubation of *T. falcklandii* is still unknown, we used 13 days as it is the average incubation of *T. migratorius* (Ehrlich et al., 1988).

For the two species with largest number of nests (*E. albiceps* $n = 27$ and *Z. capensis* $n = 35$) we used generalized linear mixed models (R package lme4 v1.1.18.1; *Bates et al., 2015*), using breeding season as a random factor to control for annual differences. For the other three species (*P. patagonicus* $n = 16$, *T. falcklandii* $n = 7$, and *A. parulus* $n = 14$), sample size was insufficient for mixed model convergence. Therefore, we used generalized linear models (*R Developement Core Team, 2018*) and excluded breeding season from the analysis, which was correlated with ground predator index for these three species. Furthermore, in a prior analysis we determined that breeding season did not have a meaningful effect on DSR for any of these three species. We checked for outliers with Cook's distance, correlation among continuous variables ($r > 0.75$), and correlation among categorical variables (assessed with a $\chi^2$ test $p < 0.05$). For *T. falcklandii* there was one outlier for concealment (Cook's $D > 1$) that did not affect model results. Therefore, we conducted all the analyses with this outlier in the data. The only significant correlation among covariates was between canopy height and understory height ($r = -0.97$) for *T. falcklandii*. We included understory height in the candidate models because it would be easier to measure in the field for future studies. For *Z. capensis,* we only evaluated explanatory variables for nests that were on the ground because all three nests above the ground were successful (there was quasi-complete separation

of data points). We replaced missing values with the mean of the variable (*Acock, 2005*). Across species and variables, 2.6% of exposure periods—the time between nest visits—had missing values. All continuous variables were standardized to a mean of zero with one unit of standard deviation for analysis (*Schielzeth, 2010*).

Before we fit nest survival models for each species, we evaluated the potential for a researcher effect on DSR based on camera deployment and nest visitation. Deploying a camera and/or visiting a nest could negatively affect DSR because parents could abandon their nests due to the disturbance. To evaluate the effect of camera presence, we incorporated an indicator variable where 1 = nests with a camera for that exposure period, and 0 = nests without a camera for that exposure period. To evaluate the effect of visits on DSR we created a continuous variable of cumulative number of visits. For this, we assumed that the effect of visiting a nest was delayed (it occurred after we left the nest) and it was higher the more times we visited a nest. If either camera or visit effect were significant, we kept the variable(s) in the final model. All analyses were performed in R 3.5.1 (*R Developement Core Team, 2018*).

## RESULTS

### Nest-site selection

We located 104 nests for the five species during three breeding seasons (*E. albiceps* $n = 28$, *Z. capensis* $n = 35$, *P. patagonicus* $n = 17$, *T. falcklandii* $n = 8$, and *A. parulus* $n = 16$). Nest-site habitat characteristics varied both within and among species (Table S1). Understory cover positively influenced nest-site selection in three of the five species (*Z. capensis, P. patagonicus,* and *A. parulus*) (Fig. 1 and Table 3). The odds of a plot containing a nest of any of these three species increased by a factor of 1.03 with every 1% increase in understory cover. Conversely, this parameter negatively influenced nest-site selection for *E. albiceps*; however, its 95% CI overlapped zero (Table 3). Understory height positively influenced nest-site selection of *P. patagonicus* (Fig. 1 and Table 3). Finally, there was a weak effect of understory height and canopy height on nest-site selection of *A. parulus* (Table 3). The models provided a good fit for the data (Table S2). For *T. falcklandii*, the best model was the null model (Table 3), indicating that none of the habitat characteristics that we measured showed strong effects on nest location. For a complete list of competing nest-site selection models, see Table S3.

### Nest survival

Of the 98 nests monitored that had a known fate, 52% of them failed ($n = 51$). The success rate per species was: *E. albiceps* 33% ($n = 18$), *Z. capensis* 44% ($n = 34$), *P. patagonicus* 63% ($n = 16$), *T. falcklandii* 43% ($n = 7$), and *A. parulus* 71% ($n = 14$). Of the 51 failed nests, 71% ($n = 36$) were due to predation. However, we were unable to identify the predator for 58% ($n = 21$) of the predation events (either the nest did not have a camera, or the camera failed to capture the event). We only identified three predators in the system: *M. chimango*, *N. vison* and *Glaucidium nana*, which accounted for 13 (87%), 1 (7%), and 1 (7%) of the depredated nests where we were able to identify the predator, respectively. *Milvago chimango* mostly depredated nestlings, whereas the latter two depredated eggs. Most

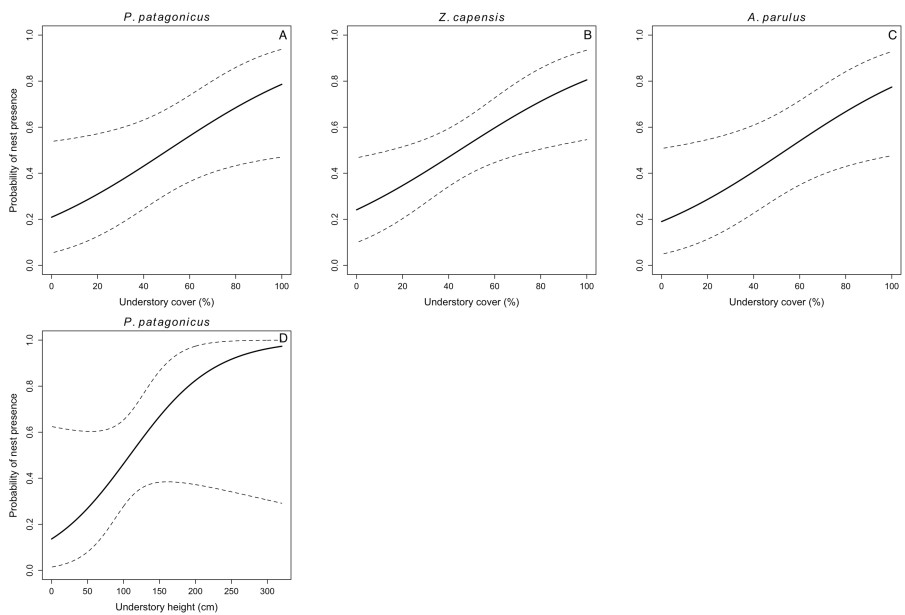

**Figure 1** **Probability of nest presence (and 95% CI) as a function of habitat characteristics for P. patagonicus (A and D), Z. capensis (B), and A. parulus (C) on Navarino Island, Chile, 2014-2017.** We only present parameters of the final model for which their CI did not overlap zero.

**Table 3** **Parameter estimates (95% confidence interval) for the best model explaining nest-site selection by five forest-nesting bird species on Navarino Island, Chile, 2014–2017.**

| Species (n) | Coefficients in the best model β (95% confidence interval) | | |
|---|---|---|---|
| | Understory cover | Understory height | Canopy height |
| *Elaenia albiceps* (22) | −0.13 (−0.298 −0.004) | | |
| *Zonotrichia capensis* (33) | 0.03 (0.007 −0.047) | | |
| *Phrygilus patagonicus* (17) | 0.03 (0.003 −0.054) | 0.02 (0.0003 −0.0423) | |
| *Turdus falcklandii* (8) | [a] | [a] | [a] |
| *Anairetes parulus* (16) | 0.03 (0.003 −0.059) | 0.02 (−0.0001 −0.0538) | −0.19 (−0.482 −0.026) |

**Notes.**
[a]The final model for this species was the null model.

predation events (69%) occurred during the nestling stage. Nest abandonment accounted for the remaining failed nests ($n = 15$). *Elaenia albiceps* and *T. falcklandii* had the highest abandonment rate, 26% ($n = 7$) and 29% ($n = 2$), respectively. In contrast, *P. patagonicus* and *A. parulus* abandoned 0% and 7% ($n = 1$) of their nests, respectively. *Zonotrichia capensis* abandoned 15% ($n = 5$) of its nests. Most abandonments (80%, $n = 12$) occurred during the incubation stage.

There was no evidence that researcher visitation affected DSR for any species. Thus, we examined the influence of temporal and habitat variables without considering the effect of our visits. *Zonotrichia capensis* was the only species where camera presence affected DSR; DSR increased by 22% when a camera was present ($\hat{\beta} = 1.97$; 95% CI [0.20–3.61]; Fig. 2). Thus, for this species only, we proceeded with model selection by including a camera

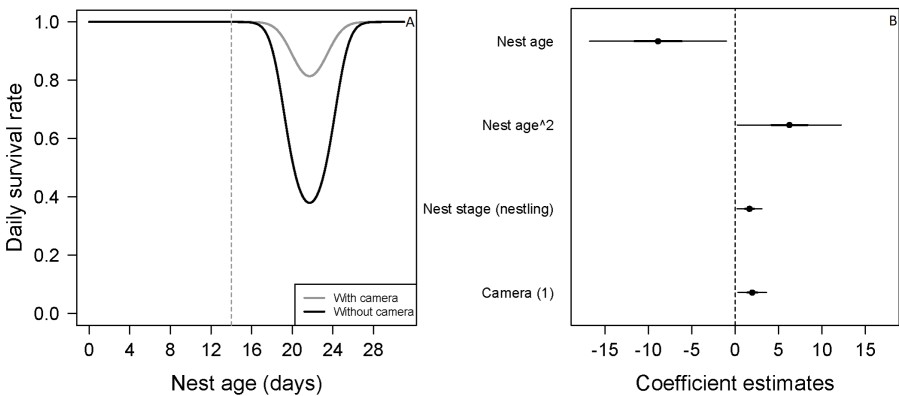

**Figure 2 Mean nest daily survival rate (DSR) of *Zonotrichia capensis*.** (A) Mean nest daily survival rate (DSR) of *Zonotrichia capensis* as a function of nest age and nest age², in the presence and absence of camera. Mean nest DSR was estimated when nest stage = nestling (1). Dashed line represents mean hatch day for this species (*Jara et al., 2019*). (B) Coefficient estimates (filled circles) for the best model ± their 95% CI (thin-outer lines) and 50% credible intervals (thick-inner lines).

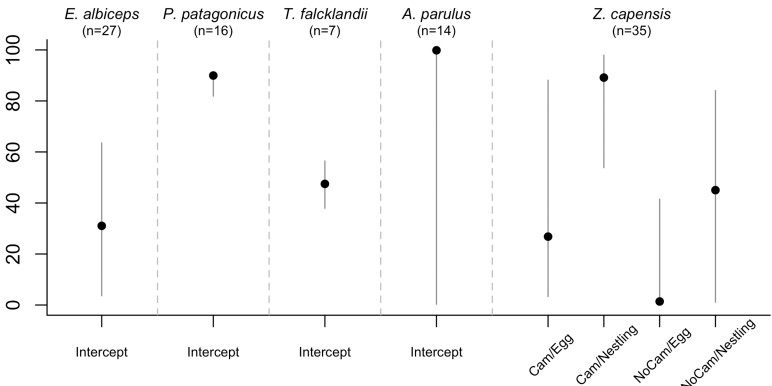

**Figure 3 Overall nest survival rate by species.** Estimated with the final DSR model for each species, holding continuous variables at their standardized mean value of 0. Therefore, for models in which all variables were continuous, the overall nest survival rate corresponds to the intercept. For *Z. capensis*, there were both continuous and categorical variables in the final model. Thus, we estimated separate survival rates for each level of the categorical variable(s) (i.e., camera presence vs camera absence, and egg vs nestling), while holding the continuous variables at their standardized mean value of 0. Once we obtained a DSR value, we raised it to an exponent representing the average number of days in the nesting cycle for each species.

effect. Overall nesting success, based on DSR, was highest for *P. patagonicus* (87.0%) and *A. parulus* (99.9%), although the latter had a large amount of uncertainty around the mean estimate (Fig. 3).

For the DSR best-supported model of *E. albiceps*, we found that there was a positive, non-linear effect of nest age, as well as a negative effect of canopy cover and understory height (Fig. 4). For *Z. capensis*, in addition to the camera effect, DSR was strongly influenced by nest age (Fig. 2). DSR followed a similar pattern in the presence or absence of a camera,

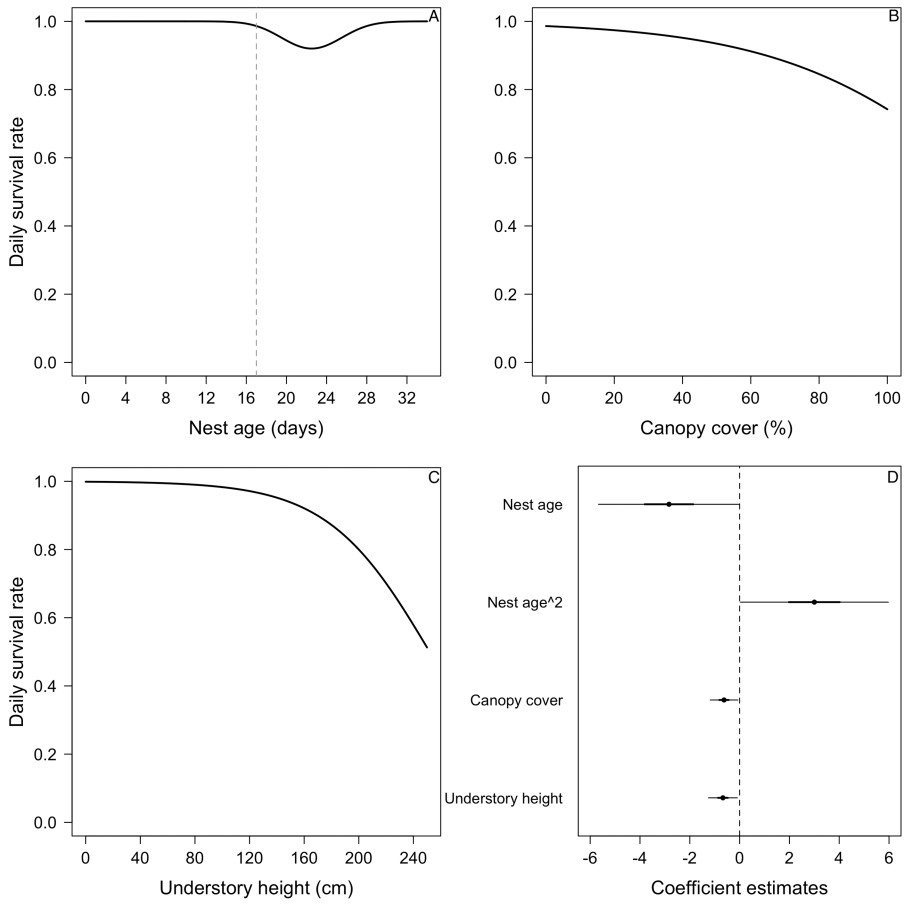

**Figure 4** **Mean nest daily survival rate (DSR) of** *Elaenia albiceps*. (A) Nest age and nest age$^2$, (B) canopy cover, and (C) understory height. Mean nest DSR for each variable estimated holding the other variables at their standardized mean value (i.e., 0). Dashed line in (A) represents the average hatch day for this species (*Jara et al., 2019*). (D) Coefficient estimates (filled circles) for the best model ± their 95% CI (thin-outer lines) and 50% credible intervals (thick-inner lines).

although survival was higher when a camera was present (Fig. 2A). Nestlings had a higher probability of surviving than eggs (Fig. 2). Overall nest survival during the egg stage (based on DSR) was 26.8% and 1.4%, in the presence and absence of cameras, respectively. Overall nest survival during the nestling stage was 89.2% and 45.0%, in the presence and absence of cameras, respectively (Fig. 3). DSR of *P. patagonicus* declined slightly with increasing nest age, understory cover, and understory height, and it strongly increased with more nest concealment (Fig. 5). For *T. falcklandii*, DSR declined with increasing understory cover (Fig. 6). Finally, we did not find any strong temporal or habitat effects on DSR for *A. parulus* (Table S4), though estimates showed a weak positive relationship with nest height and understory cover, and a positive quadratic relationship with nest age. The best-supported model for every species were good fits for the data (Table S2). For details on parameter estimates and CIs of the best-fitted nest survival models for each species, see Table S4. For a list of competing nest survival models, see Table S5.
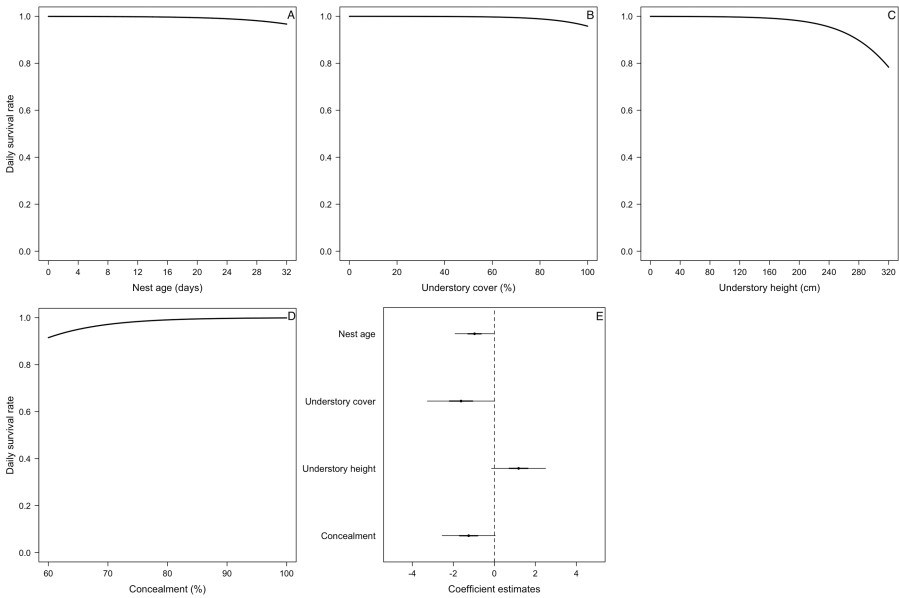

**Figure 5  Mean nest daily survival rate (DSR) of *Phrygilus patagonicus.*** (A) Nest age, (B) understory cover, (C) understory height, and (D) concealment. Mean nest DSR for each variable estimated holding the other variables to their standardized mean value (i.e., 0). (E) Coefficient estimates (filled circles) for the best model ± their 95% CI (thin-outer lines) and 50% credible intervals (thick-inner lines).

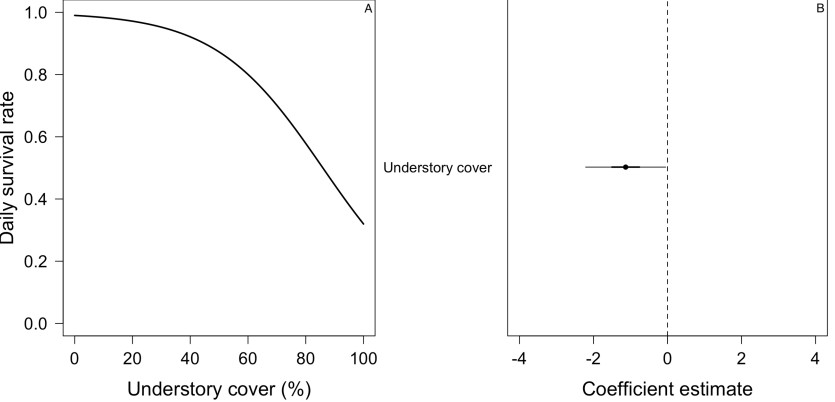

**Figure 6  Mean nest daily survival rate (DSR) of *Turdus falcklandii.*** (A) Understory cover. (B) Coefficient estimate (filled circle) for the best model ± its 95% CI (thin-outer lines) and 50% credible interval (thick-inner lines).

## DISCUSSION

Our study provides new evidence that nest survival is influenced by predation. In addition to supporting previous studies that have found similar effects (*Nice, 1957*; *Ricklefs, 1969*; *Liebezeit & George, 2002*; *Bellamy et al., 2018*; *Reidy & Thompson, 2018*), we propose a novel hypothesis that combines characteristics of the *habitat* where the nest is built (canopy height, canopy cover, understory height, understory cover—total-foliage hypothesis)
and of the *habits* of potential and actual predators (attack mode of aerial vs terrestrial predators—predator proximity hypothesis) that influence (a) nest-site selection and/or (b) nest survival (breeding success). In several cases, our sample sizes are limited, providing DSR estimates with considerable uncertainty. Therefore, our results should be taken as preliminary findings that represent baseline data for most of these species and, as a whole, provide support for both nest placement hypotheses.

### (a) Nest-site selection

We found percentage of understory cover was the most important habitat variable explaining nest-site selection, as it affected three of the five species. *Zonotrichia capensis*, *P. patagonicus* and *A. parulus* significantly preferred nesting sites with greater percentage of understory cover (Fig. 1). In addition, *P. patagonicus* preferred to nest in sites with taller understory (Fig. 1). These findings are consistent with the total-foliage hypothesis, which assumes that more foliage reduces the risk of depredation because it interferes with visual, auditory, and olfactory cues for avian and mammalian nest predators (*Martin & Roper, 1988*; *Martin, 1993*). Our findings for these three species are also consistent with previous studies on North American passerines, where species placed their nests in higher understory density and/or cover compared to non-nest random plots (*Liebezeit & George, 2002*; *Benson et al., 2009*; *Wynia, 2013*).

The other two passerine species, *E. albiceps* and *T. falcklandii*, selected nest sites with characteristics that were no different from random-paired plots. There are at least two possible explanations for this lack of effect. First, there is a pattern(s) that we were unable detect, perhaps due to limited sample size. Among the five species studied, *E. albiceps* and *T. falcklandii* exhibit the highest diversity of substrates used for nesting (*Jara et al., 2019*). In addition, *E. albiceps* and *T. falcklandii* nest at the high and low extremes of the vertical forest profile (*Jara et al., 2019*). The high heterogeneity of nesting substrate and position in the vertical axis of the forests could make it more complex to detect a pattern. Second, birds could be using an unstructured pattern for nest placement to deter predators from learning to scan for nests. This has been suggested for Hermit Thrush (*Catharus guttatu*; *Martin & Roper, 1988*) and White-tailed Ptarmigan in North America (*Lagopus leucurus*; *Wiebe & Martin, 1998*). This mechanism could also provide an explanation for the lack of significant association between nest-sites and the other two examined habitat variables (canopy height, canopy cover) in the five studied passerine species. More research with larger sample sizes is needed to elucidate this potential explanation for predator avoidance.

### (b) Nest survival
#### *Overall nest survival*

Overall nest survival rates were high for *P. patagonicus* (87.0%) and *A. parulus* (99.9%) (Fig. 3). Nest survival rate recorded for *T. falcklandii* (48%) in the remote sub-Antarctic forests on Navarino Island is higher than the 20% that has been recorded for conspecific populations in temperate forests farther north in southwestern Patagonia on Chiloé Island (42°S), Chile (*Willson et al., 2014*). In contrast, survival rates were low for *E. albiceps* (31.0%) and *Z. capensis* (1.4–89.2% depending on camera presence and nest stage; Fig. 3). The low rates we found for these two species are similar to the rates found for conspecific

populations of *E. albiceps* breeding on Chiloé Island, Chile (27% nest success) (*Willson et al., 2014*), and of *Z. capensis* breeding on central Monte Desert, Argentina (34°S; 9.4% nest success) (*Mezquida & Marone, 2001*).

On Navarino Island, *T. falcklandii* builds its nests closer to the ground than farther north in the temperate forest biome (*Jara et al., 2019*). This could be associated with the fact that mammalian ground predators are present in temperate forests, but until recently, were absent on Navarino Island (*Jara et al., 2019*). Our results thus open the following new questions regarding survival rate: (1) Could the difference in proximity to prevailing predators explain the differences in nest-placement and nest survival in *T. falcklandii* at different latitudes? (2) Can the historical absence of ground mammalian predators and the presence of aerial bird predators on Navarino Island explain the high survival rates of *P. patagonicus* and *A. parulus*? (3) Why do the nest-survival rates of these species differ from the low rates detected for *E. albiceps* and *Z. capensis*?

### Nest predators

The main cause of nest failure (71%), regardless of species, was predation. This is consistent with passerines breeding in northern hemisphere forests (*Ricklefs, 1969*; *Murphy, 1983*; *Martin, 1993*; *Wilson & Cooper, 1998*; *Duguay, Wood & Nichols, 2001*; *Liebezeit & George, 2002*; *Wesolowski & Tomialojc, 2005*). On Navarino Island, the native raptor *M. chimango* was the most common predator, accounting for 87% of the depredated nests where we were able to identify the predator, corresponding with previous studies on this island (*Ibarra, 2007*; *Schüttler et al., 2009*; *Maley et al., 2011*; *Crego, 2017*). On Chiloé Island, farther north within the south-temperate rainforest, *M. chimango* is also the main predator of passerine nests (*Willson et al., 2001*).

*Milvago chimango* is a common raptor in southern South America that inhabits a variety of habitat types, including forests, shrub-lands, steppes, and coastal ecosystems, as well as anthropogenic habitats such as plantations and cities (*Rozzi et al., 1996*). This opportunistic raptor is a generalist predator that uses a wide variety of foraging techniques. It can fish using a 'glide-hover' technique, catch fleeing insects while flying through fires, or wade to catch frogs and tadpoles (*Del Hoyo, Elliot & Sargatal, 1994*; *Sazima & Olmos, 2009*). In the forests of Navarino Island, it mostly searches for prey while perched and flying overhead (RF Jara and RD Crego, pers. obs., 2015). On Navarino Island *M. chimango* also depredates nests irrespective of their height from the ground (*Crego, 2017*). Consequently, it exerts a predation pressure from above (like other raptors) and from below (like ground predators). This suggests that birds on this island may have already developed nesting strategies to avoid ground predation pressure, even before mammalian ground predators were introduced. *Milvago chimango* populations increase with human disturbance, like those generated on Navarino Island during the last couple of decades by the king-crab industry dumping large quantities of shellfish exoskeletons. Thus, it is possible that this raptor's population has increased on the study site over time, which would represent an ecological trap, because birds on this island evolved under different historical and current predator abundance conditions (*Chalfoun & Schmidt, 2012*). We therefore recommend monitoring population

growth and subsequent impact of *M. chimango* on nesting passerines in the Cape Horn Biosphere Reserve.

The only ground mammalian predator we identified was *N. vison*, which depredated 7% of nests with a known predator. This semi-aquatic mustelid was introduced to Navarino Island at the end of the 20th century (*Rozzi & Sherriffs, 2003*) and is known for its negative impacts on native birds on Navarino Island (*Schüttler, Cárcamo & Rozzi, 2008*; *Schüttler et al., 2009*; *Maley et al., 2011*), and worldwide (*Ferreras & Macdonald, 1999*; *Nordström & Korpimäki, 2004*; *Bonesi & Palazon, 2007*; *Brzeziński et al., 2012*). However, and contrary to our expectations, its nest depredation rate on passerines was very low. A possible explanation could be a mismatch between the periods of *N. vison's* peak activity in the forest (summer) (*Crego, 2017*) and the onset of passerine nesting season (Spring) (*Jara et al., 2019*). Alternatively, because we were unable to identify the predator in 58% of the events, we may have underestimated the effect of this mustelid—and other potential predators such as feral cats and dogs—on nest survival, as birds are part of *N. vison* and cat diets (*Schüttler, Cárcamo & Rozzi, 2008*; *Schüttler, Saavedra-Aracena & Jiménez, 2018*). Contrary to previous findings on artificial nests (*Willson et al., 2001*; *Maley et al., 2011*), we found no evidence of nest predation by rodents or House Wrens (*Troglodytes aedon*). This may be because in our study of natural nests, parents can actively deter rodents and/or House Wrens (Jara et al., in prep).

### Habitat and temporal effects on nest survival: support for nest placement hypotheses

Nest-site selection was positively influenced by higher percentage of understory cover (*Z. capensis, P. patagonicus,* and *A. parulus*) and taller understory (*P. patagonicus*) (Table 3). However, for *Z. capensis* and *A. parulus*, understory cover and understory height did not affect nest survival. Furthermore, for *P. patagonicus,* these two habitat characteristics had an opposite effect, negatively influencing DSR (Figs. 4C, 4B, 5C, and 6A). Thus, it seems that these species may be selecting nest-sites that not only have a neutral effect on nest survival, but actually decrease their survival rates. Given that predation was the main cause of nest failure, it is possible that there is a disconnect between birds assessing the risk of predation (and selecting the appropriate nest-site) and the actual risk of predation. This again might be due to the above-mentioned ecological trap regarding the increased abundance of *M. chimango* due to anthropogenic factors. Furthermore, passerine populations on this island have evolved with a different predator assemblage (i.e., only aerial predators), but this has been disrupted with the introduction of exotic mammalian ground predators to this island, and the rapid increase of feral domestic cats and dogs, less than 20 years ago. This ecological trap would imply a delay in the ability of birds to adapt nesting behavior in response to a new type and/or abundance of predators. Alternatively, the mismatch between nest-site selection and DSR also could be due to methodological problems (e.g., limited sample size, wrong choice of habitat variables, etc.), or ecological-evolutionary reasons (e.g., tradeoffs with other selection pressures such as microclimate and access to food, etc.) (reviewed by *Chalfoun & Schmidt, 2012*). Further research will be needed to assess hypotheses that could explain this mismatch between nest-site selection and nest survival.

For two of the three species in which we found a nest age effect on DSR (*E. albiceps*, *Z. capensis*, and *P. patagonicus*), the pattern was similar (i.e., quadratic effect) even though its magnitude varied considerably (Figs. 2A and 4A). The low rates of nest failure during the laying and incubation periods suggest marginal effects of nest abandonment and depredation by *M. chimango* and *N. vison* (the only two identified predators during these nest stages) during the first half of the nesting cycle. Daily survival rates were lowest soon after hatching (Figs. 2A and 4A). This may reflect the sudden increase in cues to predators coming from nestlings (visual, auditory, and olfactory) and parents (visual and auditory, as their nest visitation frequency suddenly rises) (*Cresswell, 1997*; *Martin, Scott & Menge, 2000*; *Grant et al., 2005*), which increases their vulnerability to predation. After reaching its lowest rate after hatching, nest survival increased steadily during the nestling period (Figs. 2A and 4A). This pattern, which has previously been observed in passerines (*Pietz & Granfors, 2000*; *Grant et al., 2005*), could be due to increased parental nest defense as nestlings get closer to fledging (*Montgomerie & Weatherhead, 1988*). This is particularly relevant for these five species on Navarino Island, as they only have one brood per breeding season (*Jara et al., 2019*), and therefore have a greater incentive to protect their nest as young near fledging. Another non-exclusive explanation includes 'forced-fledging' of nestlings by potential predators (*Pietz & Granfors, 2000*). Nestlings that are close to fledging age may avoid depredation by leaving the nest prematurely when they are at imminent risk. This behavior may decrease depredation-induced nest failures towards the end of the nesting cycle.

Higher nest concealment for *P. patagonicus* increased its nesting success (Fig. 5D), which is consistent with the total-foliage hypothesis (*Martin & Roper, 1988*; *Martin, 1993*). According to this hypothesis, predators have a harder time locating nests with higher concealment, because it may be harder to detect them visually, aurally, and/or olfactorily. It has been suggested that *M. chimango* can detect nests visually (*Crego, 2017*), so it seems *P. patagonicus* may be trying to avoid being detected by nest predators in this system. The parental behavior of this passerine may also be an important contributing factor. *Phrygilus patagonicus* sits still on the nest in response to the presence of a predator, unlike what we observed for the other species, which flush considerably sooner and exhibit alarm behaviors (RF Jara, pers. obs., 2015). In the other species, higher nest concealment may not improve nesting success due to their more agitated parental behavior that, in contrast to *P. patagonicus,* may counteract any concealment advantage.

We found that higher percentage of canopy cover above nests of *E. albiceps* decreased their nest survival (Fig. 4B). This is consistent with the predator proximity hypothesis where nests at higher risk of predation (i.e., aerial or ground) should have lower survival. More canopy cover allows for the presence of *M. chimango*, the most common nest predator we were able to identify, because this forest raptor not only nests in the canopy, but also uses branches in the canopy to perch and look for prey (RF Jara and RD Crego, pers. obs.).

### Camera effect on nest survival

We found evidence that for *Z. capensis*, the presence of a camera increased DSR by 22%. This positive camera effect has been reported for other bird species or systems (*Thompson*

*III, Dijak & Burhans, 1999*; *Buler & Hamilton, 2000*; *Pietz & Granfors, 2000*; *Small, 2005*, reviewed by *Richardson, Gardali & Jenkins, 2009*). Cameras may have a deterrent effect on predators, possibly through neophobia towards these devices, which would consequently reduce depredation of these nests. However, this is unlikely to be the case for our study system because *M. chimango* was the main nest predator across all five species, but we only found a camera effect for *Z. capensis*. This suggests *M. chimango* did not exhibit neophobia towards the cameras. Furthermore, this raptor has been described as having low neophobia (*Biondi, Bó & Vassallo, 2010*). Alternatively, there could be a bias introduced by delaying the camera deployment until later in the nesting cycle. Nests that failed earlier in the cycle, when the camera was absent, may then positively bias our estimates of DSR for nests with a camera later in the cycle. Finally, and possibly a more likely explanation, this result was an artifact of limited exposure periods without cameras (i.e., 7.7%; $n = 83/1077$).

## CONCLUSIONS

This study provides the first data on nest-site selection and survival of open-cup-nesting passerines in sub-Antarctic forests. We also propose a novel hypothesis that represents a relationship between the habitat and type of predators. Although our study was conducted on a single location, this hypothesis could be tested for nest-site selection and nest survival in other regions. The bird species we studied selected nest-sites with more understory cover and taller understory, which according to the total-foliage hypothesis would provide more concealment against predators. However, more understory cover and taller understory decreased nest survival. There seems to be a disconnect between birds assessing the risk of predation (and selecting the appropriate nest-site) and the actual risk of predation, resulting in birds selecting riskier sites for nesting. This could be attributed to an ecological trap, where birds on this island evolved with a different predator assemblage, which has been disrupted with the introduction of exotic ground mammal predators to this island and/or the increased abundance of native *M. chimango* associated with urban development. Further research with larger sample size will be needed to assess hypotheses that could explain this mismatch between nest-site selection and nest survival.

## ACKNOWLEDGEMENTS

We thank Andrew Stillman, Karen Wiebe, and one anonymous reviewer who provided very helpful comments. We thank Edwin Price and Amy Wynia for providing edits to the manuscript. We also thank Francisco Arellano, Carla Baros, Omar Barroso, Simón Castillo, Kristine De Leon, Josefina Kearns, Javiera Malebrán, Stephanie Pincheira, Marcela Rojas, Amanda Savage, Javiera Urrutia, Ana María Venegas, Justin Williams, and Amy Wynia for their help in the field.

### Funding

Funding was provided by the Chilean National Commission for Scientific and Technological Research (CONICYT) in the form of graduate scholarship for Rocio F. Jara, the University of North Texas, Partners of the Americas fellowship, and the Agencia Nacional de Investigación de Chile (ANID, Basal Funding AFB170008) to the Institute of Ecology and Biodiversity (IEB-Chile). The funders had no role in study design, data collection and analysis, decision to publish, or preparation of the manuscript.

### Grant Disclosures

The following grant information was disclosed by the authors:
Chilean National Commission for Scientific and Technological Research (CONICYT).
University of North Texas, Partners of the Americas fellowship, and the Agencia Nacional de Investigación de Chile: AFB170008.

### Competing Interests

The authors declare there are no competing interests.

### Author Contributions

- Rocío Fernanda Jara conceived and designed the experiments, performed the experiments, analyzed the data, prepared figures and/or tables, authored or reviewed drafts of the paper, and approved the final draft.
- Ramiro Daniel Crego conceived and designed the experiments, performed the experiments, authored or reviewed drafts of the paper, and approved the final draft.
- Michael David Samuel analyzed the data, authored or reviewed drafts of the paper, and approved the final draft.
- Ricardo Rozzi and Jaime Enrique Jiménez conceived and designed the experiments, authored or reviewed drafts of the paper, and approved the final draft.

### Data Availability

The raw data are available in the Supplemental Files.

### Supplemental Information

Supplemental information for this article can be found online at http://dx.doi.org/10.7717/peerj.9892#supplemental-information.

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
