# Peer review of "Nest-site selection and breeding success of passerines in the world’s southernmost forests"

_PeerJ, doi:10.7717/peerj.9892_

## Round 0.1 · original submission · Major Revisions

We have received three detailed reviews for your manuscript. All reviewers found merits in your study, but also raised a number of issues that deserve further attention. In particular, several sections of the manuscript require a number of clarifications.

First, the hypotheses and predictions could be clearer and the introduction and discussion of the paper could be put into a broader ecological context. The methods and the results sections also require a number of changes and clarifications - see for instance the specific comments and suggestions made by the reviewers on the choice of fixed vs random effects, on randomly generating nest height for the paired sampling plots and on the modelling framework in general. The presentation of tables and figures needs to be improved.

Note that reviewers also provided several other very constructive comments and suggestions that should be integrated in the next version of the manuscript.

·

Basic reporting

Generally of high quality. I mention a couple of instances in my detailed comments below where additional citations should be added to back up some of the ideas in the paper.

Experimental design

no comment

Validity of the findings

generally good. See detailed comments where the claim that "Milvago was the main predator" needs a caveat

Additional comments

Here the authors present data on the nest site selection and nest success of 5 passerine species in a southern forest. The paper is well-written and the data analysis seems thorough. Although the sample size of nests is small for a couple of the species, there is little information currently on the reproductive ecology of such species in a southern forest so the findings stand to make a useful contribution. As a result, most of my comments are minor:

1) lines 86-89. Although I understand where these predictions about habitat and predation-risk from aerial predators are coming from, the eyesight of raptors is very keen and so it seems unrealistic, (does it not?) to think that the height of the canopy (trees) will be a significant limitation for raptors detecting prey below them? What is the actual variation in canopy height in your forest system and do you think this model effect is biologically real?

I was trying to get an appreciation for the effect of canopy height in the models and noticed in Figure 4 that the predicted plotted curve goes from a canopy height of 0 on the figure and rises quickly to a maximum nest survival at a height of just over 4 m. This seems quite strange to me--certainly the average canopy tree is taller than 4m?? Yet the effect of "canopy height" seems to be operating mainly from 0-4m high which does not seem to be biologically realistic. And within this short distance it seems unrealistic to assume that the vision of raptors is going to be limited by distance? I'd check the raw data to see whether a few odd points of canopy height might be affecting these results. In figure 4, it does not make much sense to plot "predicted" curves at levels of the independent variable that are not possible in nature. At least the predictions and discussion with respect to canopy height should be better justified.

2) the predatory kite (Milvago) seems to be a generalist predator? It certainly can hunt by perch-and wait but when I was in Chile, I often observed these birds foraging from the ground, much like corvids. Is it possible they could also depredate nests from the ground? For background, I would cite some literature on the hunting techniques of Milvago.. Also, did you ever see this raptor on the ground in your forest? Perhaps the placement of passerine nests is an "average optimum" of the different hunting techniques of Milvago--e.g., if the raptor sometimes hunts from the ground and sometimes from above, the birds may have had to balance predation risk from above and below even before the introduction of novel ground predators. This possibility should at least be mentioned.

3) Because the species of predator could not be identified in 58% of cases, you can NOT say that Milvago was the main predator in your study! You can only say that it was the most common predator of the ones you were able to identify. You do make this distinction correctly at one point in the discussion section, but you need to change the claims you make repeatedly, elsewhere in the paper. For example, line 52 in the Abstract should read "Milvago was the most common predator among the 42% we could identify. Similarly, line 381 in the discussion needs to be reworded.

4) line 278. To me, presenting these predation rates as fractions is confusing. Why not present these as straight %success and then give the sample size of nests for each species in parentheses.

5) line 432. At what distance do the parents flush when a predator approaches? Nest concealment could still be of benefit if a parent sits tight until the predator is quite close…

6) For the species in which no habitat variables predicted nest placement, one possibility is that the birds place nests "randomly" to avoid predators forming search images for nests. This seems to be the case for white-tailed ptarmigan and other grouse, for example (see Wiebe and Martin 1988, Animal Behavior 56, 1137–1144).

Reviewer 2 ·

Basic reporting

The authors examined two hypotheses regarding nest placement and its consequences for nest survival. This study is motivated by a recent introduction of ground predators to an island where the only native predators were raptors; thus, the study sought to identify nest predators, nest-site characteristics, and whether passerines selected nest sites that provided protection from both the native, aerial and introduced ground predator species.

Although the manuscript is clearly written and well-referenced, it is lacking context beyond this system. The authors can improve the manuscript by emphasizing how their hypotheses (nest concealment and predator proximity) fit into the context of general ecology to demonstrate the knowledge gap that exists outside this system in both the introduction and discussion sections. An additional shortcoming is that only one nest was confirmed to be depredated by a ground predator, which limits the scope of inferences the researchers can make about the risk of predation due to the recent introduction of mammalian predators. However, the authors have done a good job in the discussion to explain the caveats surrounding the interpretation of the depredation summary statistics.

The authors have described an interesting system and pose an important question that is relevant to this system but has implications for understanding nest placement generally. The analyses are well-written and appropriately applied. The raw data are shared but may require metadata to describe column headings and field entries; although, this is generally obvious. The manuscript can be improved by providing greater background for the hypotheses and placing the study in a more general context beyond that of this system.

Experimental design

The research has a well-defined question and a knowledge gap is clearly identified. The methods are clearly written, understandable, and appropriate for the study questions. The authors may consider generating random values for unmeasured nest selection variables and including the season effect as a fixed term rather than a random effect (see general comments). I particularly appreciate tables 1 and 2 that give the justification and rationale for inclusion of model variables.

Validity of the findings

The authors have made appropriate conclusions based on their results. However, the authors could improve the manuscript by considering the implications of their results outside of the study system.

Additional comments

Introduction:
L85-89: The justification for canopy cover as part of the predator proximity hypothesis needs better explanation or terms. The use of the term “canopy cover” here makes me think that this variable fits under the total foliage hypothesis (i.e., more cover = more foliage = more nest concealment). A better description may be canopy height and extent. I see that you have used canopy cover and canopy height in the table 1; perhaps add height to your sentence in the introduction.
L90-91: What about the interaction of these two hypotheses? The combination of these two habitat features likely contributes to nest success/failure.
L90-122: These paragraphs are very system-specific. You may consider removing some of this information from the introduction and moving it to the study site section of the methods.
L105-122: Instead of focusing only on evidence of predation pressure as a driver of nest-site selection in this system, you should present some general support for this hypothesis from other systems.
Methods:
L159-160: Does this mean that the cameras took three pictures with a minute between each picture whenever the camera was triggered? If so, a minute seems like a long delay; did you have any issues with missing what triggered the camera?
L189-196: Did you consider randomly generating a nest height for the paired sampling plots from a range of possible nest heights (could also do the same for concealment: 0-100%)? This could add a potentially important conclusion to your results and implications.
Statistical analyses: Good, clear description of analyses.
L231-233: Three levels is pretty low for a random effect and is likely unnecessary here (see Harrison et al. 2018. A brief introduction to mixed effects modelling and multi-model inference in ecology. PeerJ. 6: e4794. Doi: 10.7717/peerj.4794). Did you try treating year as a categorical fixed-effect term?
Results:
L277-279: The presentation of success/failure rate is not very intuitive. Why not present % successful or % failed?
L281-282: Do you think the failure to capture depredation events was due to the one-minute lag between pictures?
Discussion:
General: The discussion section is very system specific. What implications do your results have for our understanding of nest-site selection and nest survival broadly?
L333-334: It would be stronger to open with your general hypotheses and why that is interesting in the context of general ecology.
L335: word choice: “predicting” substitute with “correlated with”
L344: E. albiceps and T. falcklandii selected nest sites with characteristics that were no different than in the randomly sampled paired plots (they did not randomly select their nest sites).
L351-364: This section seems like it belongs within the nest survival section. The flow of the discussion would be smoother if this was moved elsewhere.
L406: abbreviate daily survival rate to DSR.
L466-468: The hypotheses are not necessarily mutually exclusive-nest concealment and predator proximity are likely interactive features of the environment that together contribute to nest placement and nest survival. This could also make it difficult to find support for one hypothesis over another.
Tables and Figures:
Table 1 is a nice addition for providing an understanding of your reasoning. Well done.
Table 2: It would be good to add the rationale for the last row of variables.
Table 5: Consider reorganizing this table in a way that allows you to easily see what variables are shared across species and how the estimates compare. Perhaps you could have the species as the columns and the variables listed as the rows, and the cell values could be the estimates, or something to that effect that makes reading the table a bit more systematic.

·

Basic reporting

1. This paper is professional and the introduction is well-written. Appropriate context is supplied in the introduction to provide a theoretical framework for the study. The authors make appropriate use of references to give background information and discuss their results in a broader context.

2. One major concern centers around the clarity and presentation of the results section. Please see notes in the general comments below.

3. I also have suggestions for improving tables and figures, included in the general comments below. Please consider these comments carefully – in their current state, I do not think the tables and figures do an appropriate job conveying the results and incorporating modeling uncertainty.

4. I encourage the authors to give a close read for occasional small typos. E.g., line 56 “not”, grammar on line 472.

5. The authors switch back and forth between scientific names and common names in figures, tables, and text. Please decide on one and use it consistently.

6. Please include a .README or Metadata file defining the fields in the raw data.

Experimental design

1. I commend the authors for gathering an extensive dataset that provides valuable insight into the breeding biology of birds in a remote and under-studied area. The authors used an appropriate modeling framework to analyze their data.

2. My major concern with the modeling framework centers around the model selection methods for survival models.
• In general, the model selection methods and results lack clarity. I suggest continued work on this section to make sure that readers can clearly understand the step-by step process that you used to build the final model.
• First, for stage 1, it is not clear in the text how the authors build the candidate model set for temporal variables. It appears from the supplement that they ran all possible combinations of variables. If so, please state this in the text.
• Second, there is an inconsistent use of AIC between survival models and nest selection models. For nest selection models, the top model was chosen based on the smaller number of parameters. For stage 1 in the nest survival models, there is no information presented on how the top model was selected. For stage 2 in the nest survival models, the top model was chosen based on having the highest number of significant parameters. This inconsistent use of AIC selection criteria introduces inappropriate subjectivity into the model selection framework, and it needs to be corrected in the revised manuscript. Please select one criteria for selecting the best model when there are multiple models within 2 delta AIC, and use this same criteria throughout.
• Relatedly, the inconsistent use of AIC also creates problems in lines 305-309. This is currently worded in a way that suggests a form of data dredging – i.e. bending selection criteria in an attempt to include variables that appear statistically significant after the model is run. There needs to be one selection criteria for the “best model”, applied across all models in this analysis.
• In lines 206-207, the authors select a top model based on the fewest number of parameters. If the revised manuscript continues to use this approach, please provide evidence for why this is an appropriate method. Generally, when multiple models are within 2 AIC, researchers choose to use the model with the lowest AIC. Even if this model has more parameters, the fact that AIC remains low means that the addition of the extra parameter overcomes the AIC penalty for parameters (-2k) and significantly increases the likelihood. Thus, it may not be appropriate to only use the model with the fewest number of parameters when it is not the lowest AIC.

Validity of the findings

1. The authors do an appropriate job interpreting their findings in the context of the results. The authors provide evidence to support both the total-foliage and the predator proximity hypothesis.

2. The authors speculate that introduced mammal predators may contribute to a disconnect between nest site selection and survival. I agree that this is a reasonable hypothesis, but there are many other potential explanations for this phenomenon. In fact, it’s actually quite common for nest ecology studies to see this trend. Please add text to the discussion to reflect this. I suggest reading and referencing the following paper in the discussion section, which reviews potential hypotheses for disconnects between nest site selection and nest survival:

Chalfoun, A. D., and K. A. Schmidt (2012). Adaptive breeding-habitat selection: is it for the birds? The Auk 129:589–599.

Additional comments

I have completed my review of Nest-site selection and breeding success of passerines in the world’s southernmost forest. I found this paper novel and interesting, and I believe the results are important to avian biology. I have included general comments below, organized by manuscript section.

Introduction

1. The introduction sets up two hypotheses operating on avian breeding biology in this system.
• The “total-foliage” hypothesis sounds very similar to the “nest-concealment” hypothesis that I see more often in the literature. Please provide text to distinguish how these hypotheses are different, or if they are the same.
• Line 42: As currently worded, this sentence indicates that the two hypotheses exclude each other. Please clarify that they are non-exclusive, and perhaps even complimentary.
• Suggested citation for framing in the introduction. This study experimentally manipulates nest height and examines predation risk.
Latif, Q. S., S. K. Heath, and J. T. Rotenberry (2012). How avian nest site selection responds to predation risk: testing an ‘adaptive peak hypothesis.’ Journal of Animal Ecology 81:127–138.


2. In order to support the claims in the introduction, more detail needs to be provide on the differences in the predator communities between the island and the mainland. Is there evidence that mammal predation is more important than raptor predation on the mainland?
• Line 108-109: How do we know that this difference is not just due to differences in vegetation type or vegetation height?

3. Line 129-130: If the authors hypothesize that nest height has a quadratic relationship with survival, why isn’t nest height tested as a quadratic variable in the candidate models?

Methods

1. Please see the concerns in the experimental design section above.

2. Have you considered using a random slopes model to make population-level inference? This would allow you to include all species in a single model, with a random slope effect for species. For the nest selection model, for example, this approach would estimate a population-level effect of understory cover, as well as a unique estimate of understory cover for each species. Random slopes models allow species with small sample sizes to “borrow strength” from species with larger sample sizes.

3. In table 2 and throughout the paper, I suggest replacing “biological and temporal” with just “temporal”. The current wording is confusing because habitat variables are also biological.

4. Line 245-246. I don’t understand why the choice was made to exclude this variable. If all nests above the ground were successful, this indicates a strong effect of nest height on survival.

5. Line 200: Please clarify that a separate model was run for each species. This is currently unclear.

Results

1. Overall, I believe the results need to be substantially edited for clarity. The results are highly repetitive, inefficient, and difficult to follow.
• Please remove areas where the results are simply repeating estimates that are already reported in tables (e.g. lines 297-299).
• Lines 297-298: it is unclear where these numbers come from, and why some a ranges and others are not. Report these numbers in table 4, and describe overall relationships in the text. Please include CIs for all estimates of total nesting success.

2. Table 3: The intercept estimate for Austral Thrush should be reported as 0.

3. Table 4: The organization of this table is unclear. Why is the camera effect split into egg and nestling stage, and why are estimates only presented for camera=absent in the Austral Thrush?

4. Table 5: I suggest changing this table so that it has a column with coefficient estimates (with CIs included), rather than writing out the model formula. If there is a random effect, include the estimated standard deviation of the random effect distribution.
• Please note that the notation used in this table is not intuitive for people who do not have experience using glm()-class functions in R.

5. Figures 2-6.
• It is not necessary to give the model equation in the figure caption.
• Please include confidence intervals on all line plots.
• If table 5 is changed to include CIs and coefficient estimates, the coefficient estimate figure will no longer be necessary.
• Why is a line splitting up nest stages included for all nest age plots, except for figure 5? In figure 3 and 5, why aren’t the estimates for nest stage presented as a figure. Effects of categorical variables on DSR can easily be presented as boxplots.
• To help clean up the figures, I suggest only including figures for statistically significant parameter estimates. Readers will be able to reference the table to see parameters estimates for non-significant effects in the top model.

6. Line 270: Replace “increased by 1.03” with “increase by a factor of 1.03”.

7. Line 278: Replace success/failure rates with either successes/total or failures/total, so that results are more interpretable and match the reported rate on line 277.

8. Supplement
• Please include captions for all tables in the supplement.
• What are the little cross symbols next to variables in supplemental table 3?


Discussion

1. Line 344: The claim that nest selection was random is not supported by the results (e.g. there could be unmeasured variation that the birds respond to). Instead, the authors can make the claim that among the habitat variables that they measured, none had a strong effect on nest site placement.

---

## Round 0.2 · Minor Revisions

We have received two reviews from reviewers who previously reviewed your manuscript. Both felt that the manuscript was much improved. However, they also suggested a number of additional changes that deserve further revision. In particular, some sections of the results and discussion require some more work to make them clearer. Reviewer 3 (Andrew Stillman) also identified specific concerns regarding the daily nest survival rate (DSR) and your sample size and he made specific suggestions to address them. Please integrate convincingly all of his suggestions related to this important aspect in your next revision, as well as others comments provided by both reviewers.

Reviewer 2 ·

Basic reporting

The authors have improved the clarity of their writing in the revised draft. The addition of examples from other species to the introduction and discussion has increased the ecological scope of the manuscript.

Experimental design

The context of the study and research questions are well defined. The methods have been improved since the previous draft and are more clear in this version. The tables and figures are also clear. Well done!

Validity of the findings

The authors have done a good job incorporating examples from outside of their study system that increase the ecological scope of the inferences. I appreciate the additions the authors have made to the discussion; however, some sections are quite long. It may be possible to edit some of the text down. I also suggest considering the addition of some subheadings to the Nest Survival section to help structure that section a bit more.

Additional comments

L78-79: Can you briefly explain what you mean by "simpler structure" in high-lat vs. tropical forests? Some readers may be unfamiliar with the structure of these forest types.

L326: comma needed after "In addition"

L327: "effect" should be plural

L360-482: This section is long with lots of references to analyses you conducted. Consider adding some subheadings to this section that correspond to the specific analysis you talk about to help cue the reader to remember the method/result you discuss.

L373-378: This is a matter of style, but the questions here read a bit informal. I would state these ideas another way for a more professional presentation.

L334, 360, 484: Make sure that the headings are consistently formatted. L334 does not match the others.

·

Basic reporting

The authors have adequately addressed my comments on the previous version, and I feel that the manuscript is much improved. I especially commend the authors for their changes to the text in the introduction and discussion, which now frame the study in a more compelling way.

Though the writing in the results section has improved, there is still some work to be done in the last paragraph of the results. Here, the results read like a choppy list that repeats the information in the figures and tables. I realize that it is difficult to find the right balance, especially when there are results for so many species. Please spend some more time on this paragraph, re-organizing it to read less like a list. Note that it is not necessary to describe every single variable in the text – it’s okay to only focus on the ones that proved important in the analysis.
Example for A parulus: “We did not find any strong temporal or habitat effects on DSR for A. parulus (Table S4), though estimates showed a weak positive relationship with nest height and understory cover, and a positive quadratic relationship with nest age.”

In addition, some typos remain. Here are some examples:
78: “Hight-latitude”
Table 1: Understory cover: change “interfere” to “interferes”
273-274: Confusing syntax “both of these parameters had their CI overlapped zero”
289: change “nest” to “nests”
326-327: Syntax error “In addition to support previous”

Experimental design

The authors have done a good job addressing my previous concerns about model selection methods.

Validity of the findings

Much improved, but please see comments in the “results” section below on unreasonable predictions for DSR values, and suggested changes.

Additional comments

I have completed my second review of Nest-site selection and breeding success of passerines in the world’s southernmost forest. Overall, the authors have done an adequate job incorporating reviewer’s suggestions, and the manuscript has substantially improved. My remaining concern (described above) centers around the need to clearly present the uncertainty associated with the low sample sizes in this study.


INTRODUCTION
No additional comments

METHODS
203: The sentence that starts with “We considered…” is redundant with the previous sentence.

231: “Two species with more nests” is confusing – it’s not clear here that you’re referring to sample size.

To make the methods more repeatable, please report the number of days in each nesting stage that were used to calculate nest survival rates from DSR. This could be reported either in the main text or a supplement.

RESULTS
Please see the comments above about suggested changes to improve the last paragraph.

I also have an overall concern about the DSR values reported in this study. Many of them are exceptionally low – lower than the realm of possibility for these data. For example, a DSR value of 1.4% (line 301) translates to essentially 0% chance of success. This may be true for the handful of sampled nests, but it should not be taken as a population-level estimate. DSR values in the plots regularly dip down to levels that render a population functionally extinct. For example, a DSR value of 0.8 over, say, a 20-day nesting cycle means that only 1% of nests actually fledge young. I see two potential reasons for this, and both must be clarified in order for these results to be trusted.

(1) The sample sizes used in this study are very low, meaning that DSR estimates are highly biased and include lots of uncertainty. These data are valuable and I think that the analysis is justified (a lot of hard work when into collecting these data!), but the authors need to openly and explicitly address this limitation, report that these estimates represent a baseline for an under-studied population, and perhaps state that more research is necessary to produce more certain estimates. The fact that certain variables are important predictors is all the more impressive given small sample sizes, but as an ornithologist I cannot place much faith in the intercept DSR estimates due to the sample size. After adding sample sizes to figure 3 (see comment below), I also suggest adding a short paragraph to the discussion section providing caution about these estimates and clarifying that they represent small-sample baselines.

(2) I’m suspicious that the x-values in figures 4, 5, and 6 extend beyond the range of values sampled. For example, in Figure 4, did canopy cover for E. albiceps nests actually range from 0 to 100? Did canopy height measured at nests range from 0 to 18? Forcing a model to predict outside of the range of fitted values will produce unreasonable DSR predictions (particularly for logistic functions!), like the extremely low predictions presented in some of these plots.

275: Please clarify: “none of the habitat characteristics that we measured showed strong effects on nest location”

300: Suggested change to wording. Instead of saying “larger CI”, I suggest using a phrase like “greater uncertainty around the mean estimate”.

301: This number is incredibly low and has a huge amount of uncertainty, probably due to a very low sample size. Thus, I don’t think it’s beneficial to report it in the Results section because it is likely a biased estimate that doesn't represent the general population. I think it’s fine to include estimates for species with low sample size in Figure 3, but sample size should be stated more clearly in the figure.

312: “Eggs’ DSR” is awkward wording.

Figure 1: In the bottom panel, the CIs look like they are predicting estimates above 1, which is impossible. Perhaps there is a problem with the back-transformation of the predictions from the logit scale? Please correct this error in the predictions (e.g., plogis() function in R).

Figure 3: I suggest placing the sample size for each estimate above the lines. It is clear that low sample sizes are severely impacting estimates here (e.g., A. parulus, Z. capensis), and placing the sample sizes in this figure will help readers understand why uncertainty is so high.
The caption for this figure is rather informal and could use some clarification. There are two sentences that end with “in it”. The last sentences could be clarified with wording like “… to an exponent representing the average number of days in the nesting cycle for each species”.

DISCUSSION
As suggested above, please include a paragraph discussing sample size limitations and how they affect interpretation of the baseline DSR estimates.

---

## Round 0.3 · accepted · Accept

I am generally satisfied by the final modifications made on the manuscript.